# The Catholic Church and Refugees in Slovenia

Srečo Dragoš 

Faculty of Social Work, University of Ljubljana, 31 Topniška, 1000 Ljubljana, Slovenia; sreco.dragos@fsd.uni-lj.si

**Abstract:** In Slovenia, the Roman Catholic Church (RCC) is the largest, most influential, richest, and most politically significant among all religious organisations. In estimating its contribution related to migration and the refugee situation it is necessary to consider the wider context of Slovenian society, primarily four characteristics: (1) despite the principle of separation of state and religious spheres, it is in relation to the Roman Catholic Church that the bulk of unresolved issues and occasional exacerbations occur; (2) while Slovenian public opinion is rather volatile in its expression of social distance to foreigners, it does not represent the main problem; (3) Slovenia's state politics are very closed to refugees; and (4) political parties from the right wing of the political spectrum are rather xenophobic. While the Catholic Caritas plays a positive role in the care of refugees, the RCC has always supported the right-wing political parties when they come to power, and it is also susceptible to Islamophobia.

**Keywords:** migrants; refugees; Church; Islamophobia; charity; social function

## 1. Introduction

In addressing the titular question—what is the religious contribution to dealing with refugees (within a national context)—it is necessary to point out five facts that, if ignored, are the most common sources of misunderstandings.

Firstly, numerous social functions[1] still performed by faiths and religions are not irreplaceable; for each of them, the more societies are modernised (functionally differentiated), the more possible non-religious and non-spiritual alternatives exist. Secondly, the functional effects of faith and religion can work in both directions; they can reinforce social cohesion as well as trigger conflict, often both simultaneously: "social unity at one level often can produce conflict on another level" (Furseth and Repstad 2006, p. 164). Thirdly, when religious interpretations become institutionalised (and turn into religions),[2] the societal effects of this cultural phenomenon intensify. Fourthly, despite the development of sociological research, it is still uncertain which of the five models of "civilizing global religious conflicts", as enumerated by sociologist Ulrich Beck,[3] is the most potentially effective. Therefore, we are still without a "correct" theory and are relying only on empirical examples of what works and what does not. Fifthly, the escalation of social conflicts seems to be proportional to the ignorance of the question of why religious history and identity so often form the core of nationalisms. These are re-emerging in the 21st century, although historical evidence provides a comprehensive, straightforward, and important answer to this question: "religion was there the first" (Bruce 2003, p. 79). Therefore, the aforementioned warnings, which must be taken into account regarding religion, also apply to nationalism.

In the assessment of the function that the Roman Catholic Church (RCC) in Slovenia has in relation to refugees, it is useful to remember two warnings ("methodological assumptions") emphasised by the classic functional sociological analysis by Robert K. Merton (1979). These are contextualisation and the distinction between functional, dysfunctional, and non-functional effects. In relation to the first emphasis, this article addresses three dimensions:

- The normative embeddedness of the RCC in the legislative system regulating the religious field in Slovenia, and the RCC's attitude toward minority religious and ethnic categories, including refugees.
- The attitude of the Slovenian public toward refugees.
- The migration policy of the Slovenian state toward refugees.

Concerning refugees, the above dimensions (normative system, public opinion, and state migration policy) are crucial for assessing the context in which the RCC operates in Slovenia. For any organisation, its specific activities can yield different results or effects in various circumstances, depending on the specific conditions in which it operates. Without understanding this specifically Slovenian context, functional analysis would remain too partial, as the role of the RCC as an intermediary institution would be underestimated. Intermediation refers to mediating between civil society and authority (Mali 1998), and among all religious institutions in Slovenia, the RCC is the most influential and powerful organisation in this sense. The contextualisation of the RCC—that is, the interdependence and entanglement of its operation with normative status, public opinion, and state policy—is well researched in Slovenia in many areas (Roter 1976; Kerševan 2005; Smrke and Hafner-Fink 2008; Smrke 2009, 2014; Šelih and Pleterski 2002; Črnič and Pogačnik 2021; Jogan 2023; for the former Yugoslavia: Ognjenović and Jozelić 2014)—except for the attitude toward refugees. Therefore, in the next section of this article, I address the connection between Slovenian legislation in the religious field and the operation of the RCC (in Merton's sense of distinguishing between different functional effects).

The third section describes charitable activity as part of the RCC's activities. The fourth section outlines one of the more negative periods of the RCC's activity in more recent history; this concerns Islamophobia, with regard to which the RCC did not take the righteous side, but poured more "oil on the fire". The fifth section calls attention to the attitude to migration of Slovenian public opinion, which despite its volatility is not the main factor in increasing social distance—the main danger is presented by state politics, as shown in the last section. The listed highlights are requisite to understanding the weight or role of the RCC in the attitude of Slovenia to refugees.

## 2. "Blind Spots" of State Regulation of the Religious Sphere in Slovenia

From the Second Vatican Council six decades ago, up until the foundation of Slovenia as an independent state (1991), the relations between the then state of Yugoslavia[4] and the RCC were exemplary at the micro as well as macro levels. At the level of everyday life in Slovenia this meant that—despite the democratic deficit of a single-party system—there were practically no religious or anti-religious conflicts among the population and even fewer incidents. At the macro level, the former Yugoslavia and the RCC made a diplomatic agreement laying down that the state ensures religious freedom, autonomy, and pluralism; that the RCC does not interfere with politics, and also "no longer insists on restoring its position and rights" that the RCC enjoyed in the pre-socialist past (Roter 1976, p. 281).

Confusion and occasional tensions between the state and the RCC, which have not ceased up to today, began after 1991 with the foundation of an independent Slovenia. At that time two main problems arose; the first being the ownership problem and the second the problem of discrimination, for which the state is responsible even more than the RCC. Namely, within the denationalisation legislation, the right to the restitution of its entire property[5], which was nationalised after the Second World War, was granted to the RCC by the new Slovenian state. The Slovenian version of denationalisation—which was 100 percent naturalisation—was the most extreme of all post-socialist states. Not only did it trigger new injustices and prolonged proprietary legal disputes between the RCC and the state (which have not been resolved to this day, three decades later), but it also enabled the RCC to become by far the richest institution among all religious as well as non-religious organisations of civil society in Slovenia. Furthermore, among all religious organisations it was the RCC that became the most abundantly financed by state resources; although it is, as aforementioned, the richest religious institution in Slovenia, as well as in the world. The

second problem regarding discrimination was exacerbated with the applicable *Freedom of Religion Act* (ZVS 2007). Built into this Act were two "blind spots" that were lobbied for by the representatives of the RCC, and the state fell for it. These two "blind spots" are as follows:

1. To provide the possibility of state funding for religious organisations, the aforementioned Act was written so as to explicitly lay down that "churches and other religious communities/…/are generally beneficial organisations" (ZVS 2007, Article 5). In reality, there is no such thing! This is a rigid ideological postulate that was refuted in sociology seven decades ago with the critique of classical functionalism proposed by the sociologist Robert K. Merton. The point of Merton's warning was that no institution, without exception—neither the family nor the state, church, school, judiciary, or stamp collection society—can be *a priori*—i.e., in advance, self-understandingly, without any empirical evidence—proclaimed as generally beneficial. Namely, all and any kind of phenomena and institutions can produce functional, *dys*functional or *non*-functional effects (the latter involves consequences "that are simply irrelevant to the system under consideration"; Merton 1979, p. 115). This means that a functional qualification of a phenomenon or an institution is clearly an empirical question, and it cannot represent an *a priori* normative judgement that would be valid *a posteriori* and *a priori*, independent of empirical facts. This warning implies an important hypothesis that should be verified at each estimation of all institutions including religious ones. It can be formulated like this: the degree of the development of an institution—its age, size, level of its structure, and ramification or diversity of its activities—is directly proportional to the possibility that this institution reproduces all three types of the aforementioned effects. Namely, the increased degree of the development of an institution also increases the diversity of its functional impact on its environment and, in turn, the likelihood that the institution functions—*at the same time*—functionally and dysfunctionally, or non-functionally, depending on the field and activity that are being estimated. Being one of the oldest, richest, and most hierarchical institutions in the world, it is inadmissible to normatively qualify the RCC as simply a "generally beneficial organisation".

2. Regarding the material support of religious organisations coming from state resources (budget), the aforementioned normative *Freedom of Religion Act* has had a discriminatory provision built into it that is still applied even today and that nobody problematises as a principle. This is the provision that curtails the equality of religious organisations, perverting it. Equality began to be practised in terms of different resources being allocated to different organisations—but in a contradictory way. Where is the trick? If the state decides to materially support religious communities, then it can do so, *and* also respect the principle of equality only in two ways: by funding all religious organisations with an equal amount of financial means or by funding them differently, because of the diversity of religious organisations. If we decide on the second option ("different resources to different organisations") then the only way for the state to practice this in a non-discriminatory way, in the name of equality, is to allocate more money to smaller religious organisations—because they have less members, so it is more difficult for them to self-finance their infrastructure, which is why they are financially weaker—compared to larger and richer ones that are able to maintain themselves. If the state does the opposite, as it actually did by adopting the Religious Freedom Act, then this no longer constitutes equality between religious organisations, but represents the blatant discrimination of smaller organisations to strengthen the monopoly of the largest. This privilege for the RCC is facilitated by a legal norm stipulating that the state only pays contributions to social security for the largest religious organisations, where the numerical ratio of believers reaches "at least 1000 members of a registered church or other religious community per one religious servant" (ZVS 2007, Article 27). If Slovenian legislation followed the constitutional principle of the separation of the religious sphere from the state, the opposite should be true. In other words, if the state wanted to financially support the pluralisation of the religious sphere, it should support smaller churches or religious communities that cannot sustain themselves due to

having a smaller number of believers per religious functionary. From the perspective of the separation between the state and religion, it is important to remember that all religious choices must be considered equal. The state should never define any of them as more or less important, especially not rank them based on their "spirituality" (which is also one of the criteria for state funding of religious organisations; *ibid*.: Article 5). Therefore, Slovenia allocates the most financial resources each year to the RCC, even though, as mentioned, it is by far the wealthiest religious organisation and can easily sustain itself (Dragoš 2014). Substantially smaller amounts of resources are allocated to the other three churches in Slovenian territory (Evangelical, Islamic, and Orthodox), which are smaller than the RCC; while all others which are very small are given nothing, despite their combined number making them the largest group.[6]

This regulation of relations between the state and religious organisations leads to latent erosion of the civilisation principle, whereby the state is considered separated from the religious sphere; this is even written in the Slovenian Constitution (URS 1991, Article 5).

### 3. Functionality of the RCC

As the RCC in Slovenia is the largest, richest, and state privileged church, it has the largest influence of all religious actors on the public. This influence is twofold. Using Merton's terminology, the RCC's influence can be defined as functional in the area of strengthening solidarity with refugees; at the same time, however, the RCC is occasionally dysfunctional in terms of spreading ethnic and religious prejudice (primarily Islamophobia). Let us first consider its first, positive function.

Regarding its attitude to refugees, the RCC's role is positive mainly due to its charitable activities within the Slovenian Caritas. Even before the first mass arrival of refugees to Slovenia and Europe in 2015, the rest of the world and the Slovenian public were worried about the violence in Iraq. According to UN estimates, 1.8 million people migrated from Iraq (due to the ISIS reign of terror). The Slovenian Caritas was one of the first humanitarian organisations that decisively called upon the Slovenian government and proposed that Slovenia receive "as many [refugees] as possible" because we have enough experience to be able to do this:

> "Can we accept them also in Slovenia as soon as possible, within a month? This is for the Slovenian government to discuss as soon as possible and adopt concrete decisions about how to enable a temporary stay in Slovenia to as many as possible. Both state institutions and humanitarian organisations have enough experience with refugees to be able to do this and send the world the message that we are people!" (ŠK—Koper 2014)

Equal appeals can also be heard from officials at the top of the RCC in Slovenia, in different periods (Glavan 2015; SŠK 2021).[7] They also abundantly refer to the expressly humanistic appeals of the current Pope Francis, emphasising among other things also what the Slovenian political elites are reluctant to hear: i.e., that the fencing in of the state border with barbed wire is no solution, because "problems are not solved and co-existence is not improved by building higher walls, but rather by employing joint forces in caring for others as best each individual can, and respect for the law, always giving priority to the inalienable value of life of every human being." (Mesojedec 2021). Both the top ranks of the RCC in Slovenia and its highest body, the Slovenian Bishops' Conference, as well as all local branch offices of the Slovenian Caritas have never, in any of their numerous press releases expressed a need to socially distance from the refugees, stirred up prejudice, or spread fear of refugees. It is this principled attitude, including concrete acts of solidarity in the field, that places the RCC in Slovenia—despite its otherwise ultra-conservative stance and pre-Council orientation—in the majority group of civil-society organisations that are strongly supportive of the refugees and that form the polar opposite to the right-wing-oriented political parties in the Slovenian Parliament that, hypocritically, keep referring to Christian values. Annual financial statements summarised in Table 1 show that the attitude of the Slovenian Caritas is not mere moralising.

**Table 1.** Slovenian Caritas—material help to refugees (SK 2023a, 2023b).

| Year | In € | % of Total Expense |
|------|------|--------------------|
| 2007 | 130,650 | 2.0 |
| 2010 | 282,803 | 3.3 |
| 2013 | 164,886 | 1.7 |
| 2015 | 614,499 | 7.7 |
| 2019 | 219,958 | 2.0 |
| 2022 | 623,947 | 3.7 |

Although nominally these are not large sums being spent on refugees, in Slovenia they are comparable with other, larger charitable organisations (such as the Red Cross of Slovenia; RKS 2023). At the same time, we need to be aware that Slovenia is among the countries with the smallest number of refugees in Europe, in absolute as well as relative terms, considering the size of its population (for more, see Section 6). Table 1 shows that, among its diverse activities, the Slovenian Caritas dedicates around three to four percent of all its material expenses to refugees; during the largest refugee influx (2015), this was even as much as 7.7 percent. It also needs to be considered that Table 1 does not even show an evaluation of the number of hours of voluntary work provided, which is substantial. The Caritas 2015 report reads that within a single week, the organisation provided "over 50 volunteers on the daily day-shift and around 20 on the night shift. Until now 257 volunteers participated on the shifts. Since Saturday the Caritas volunteers have given over 3000 voluntary hours." (SK 2015).

The only two verbal slipups that somehow ruin the portrayed image of the RCC are the following: first, the imprudent bishops' statements; and, second, the explicit appeal of the ultra-right-wing Bishop Štumpf. In the first substantial refugee wave in 2015, when visiting a refugee centre, two bishops felt the need to warn the public against the tensions arising between the refugees and the original Slovenian nation "because of the differences compared to European culture, and because of the diversity of ethnic groups (Arabs, Afghans, Pakistanis . . .) within the mass of refugees" (ŠC 2015). The public did not fall for this implicitly xenophobic "diagnosis" of the problems of the receiving environment with the refugees; the same goes for bishop Štumpf's provocative appeal. He issued an appeal that read, "What do you do with violent migrants? Immediately and unconditionally send them from where they came from." This call was immediately supported by the right-wing media, which at the same time portrayed Bishop Štumpf as a target of the left-wing media and the one who will "again receive an avalanche of insults on the part of the left-wing migrant-lovers. . ." (Nova 24TV 2019). The Slovenian public did not fall for this fanning of xenophobia either, and this statement was also not supported by the Bishops' Conference; although neither was it labelled inadequate by the top brass of the RCC. Instead, the RCC's webpage published Bishop Štumpf's explanation of what he said by saying that he had been utterly misunderstood, and how in reality he was favourable rather than negative to the refugees. He defended this image of himself with the following words:

"Individuals and groups are also coming in an organised way, and not in small numbers, who do not hide their Islamic extremism. Most of them are encouraged to be artificial migrants who abuse the status of being a refugee to be able to enter Slovenia or Europe unhindered."

"They each need to be given bread with respect and human attention regardless of their religious or ideological belief/. . ./However, if someone rejects this bread only because it is given by a Christian hand, then I know they are not hungry/. . ./In short: I do good, but I am not naive. Also, it is a bishop's duty to warn believers of the impending danger."

"For how long will the volunteers of Caritas and other humanitarian organisations have the strength to serve refugees, and also artificially encouraged migrants?"

"Our ancestors survived Turkish invasions, diverse social systems and also frequent emigration. They leaned on the Christian religion and hereditary honours. Nobody renounced themselves/.../We do care about what is happening presently in the Pomurje region. We are also worried for the whole of Slovenia."

"Slovenians cannot afford the mistakes that large European nations can. Size smooths over the mistakes more quickly. But Slovenia is small, and, therefore, each small mistake is fatal for it." (Štumpf 2015)

In the eyes of the public, these statements by the Bishop Štumpf did not threaten the positive function performed by the Slovenian Caritas within the RCC. Of course, neither did they reduce the xenophobia that within the Slovenian population had been growing even before the first refugee crisis in 2015 (Žagar 2023).

## 4. Dysfunctionality of the RCC

In terms of social impact, the functioning of the RCC (in the three decades of the independent Slovenian state) was the most harmful and resounding in three areas: in the religious–ethnic, in the economic, and in relation to sex crimes.[8] In the following I will limit myself to discussing the first area, in which during the time of the largest (intentional) fanning of Islamophobia in recent Slovenian history, the RCC was a close ally of xenophobic political parties.

This was an organised Islamophobic campaign against the efforts of Slovenian Muslims to build the first and only mosque on Slovenian territory. This aversion to the mosque has been evidenced in a latent form ever since the first initiative for its building was expressed as early as 1969, during the time of the socialist government, which is why in the next half a century there were no further developments in this regard. The result of this political ignorance escalated in 2001 in the manifest form of Islamophobia. When more concrete plans for the positioning of the mosque in Slovenian space were designed, the opponents of the building of the (first and only!) mosque launched an efficient media hate campaign, including "jumbo" posters and street demonstrations; and even launched a campaign for a referendum about the rights of the Muslim minority.[9] This manifest expression of Islamophobia went on until the beginning of the actual building of the mosque (the foundation stone was laid in 2013, and the building was completed in 2020).

To illustrate or understand the problem (its contextualisation) it needs to be underlined that until then, Slovenia was a unique—and scandalous—example on a global scale! It was the only European state without a single mosque, despite the fact that Muslims in Slovenia represent 2.4 percent of the population; in a religious sense they are the second largest religious community and ethnically they are the third largest minority in Slovenia. If the constitutional principle on the equality of religious communities were implemented consistently, Slovenian citizens with Muslim religious belief should have at their disposal no fewer than 125 mosques to reach the number of believers per a religious object as is applied to the Slovenian Catholics (i.e., 378 believers). Needless to say, nobody has ever strived for such a proportionality. The Muslims only requested one mosque in the whole of Slovenia. The building of this mosque—if it were allowed—would be entirely self-financed, that is, without any state resources. With this initiative of the Muslim community in mind, we should mention the actual number of religious objects in Slovenia. There are a single mosque (only since 2020!), three synagogues (of which only one is in use), three Orthodox churches, and around 3000 Catholic churches. In a five-year period (1998–2002), as many as 50 Catholic churches were newly built or restored in Slovenia; all of which received financial aid from the state.

The analysis of Islamophobic statements produced by the political functionaries and theologians of the RCC (Dragoš 2004) in the explicit phase of Islamophobia, shows that these statements can be classified into six general argumentation types, as shown in Table 2.

**Table 2.** Typology of Islamophobic argumentation against the building of the mosque in Slovenia (Dragoš 2004, p. 14).

| Types of Islamophobic Argumentation | Core Idea |
| --- | --- |
| 1. The zero-sum argument | If one side gains, the other necessarily loses something |
| 2. The sanitary–landscape argument | The Slovenian cultural landscape should remain uninfected by non-Slovenian foreign bodies |
| 3. The assimilation argument | A minority has to adapt to the majority, the foreigner to the native—or else the opposite occurs |
| 4. The anti-colonisation argument | Immigrants are always followed by their relatives and friends, the immigrant community thus grows continuously until it threatens the autochthonous majority (whose birth rate is falling as it is) |
| 5. The patronising argument | "It is for your own good"—listen to us, it will be for your own benefit |
| 6. The conspiracy theory | The current ruling forces and communists use the Muslims for their fight against democracy, the RCC and the Slovenian nation |

All of the types of intolerant statements from Table 2 were created in Slovenia "indigenously" even before the attack on the World Trade Center twin towers in New York on September 11 2001; this is with the exception of the sixth type of argument, which emerged after this date. Propaganda in the sense of the above "arguments" quickly turned the Slovenian public towards an Islamophobic direction (although it did not have long-term effects). In only three months, public opinion changed as follows: in December 2003, the share of those who were in favour of the building of the mosque was six percent, which was still higher than the share of those in opposition to it. However, in February 2004, there were already 27.5 percent more opponents of the mosque than advocates. Still greater resistance to the mosque was expressed by religiously self-determined respondents.[10] In the first month of polling, there were 16 percent more opponents than advocates of the mosque among the religious respondents, while in the third month the majority category of opponents increased by a further 12 percent, and the minority category of advocates additionally dropped by 37.5 percent (Dragoš 2004, pp. 25–26).

Three consequences followed this sad episode in recent Slovenian history, as follows:

1. The main actor in the fanning of Islamophobia in Slovenia was not only the clerical top of the RCC, but the conservative right-wing political parties (the very same ones that among all Slovenian political parties are the only ones to refer to Christian values, both in their principled programme as well as public statements and practical actions).[11]

2. The leaders of the Slovenian RCC, who are utterly conservative and pre-Council in their orientations (Jogan 2023; Kerševan 2005, 2011), actively supported the Islamophobic statements of some Catholic theologians of the Slovenian RCC by never distancing themselves from them or denying them; at the same time, the Slovenian RCC has continuously—i.e., in all three decades of the existence of the Slovenian state—aligned itself with the right-wing political parties that were the spearheads of Islamophobia in the outlined period.

3. This orientation of the RCC in Slovenia is indicative of Islamophobia, although sociologically and historically, it is far from surprising despite the fact that it does not have the majority support of the Slovenian public (Smrke 2009; Smrke and Hafner-Fink 2008).

### 5. The Attitude of Slovenians to Foreigners

After the first decade of the declaration of the independent state, Slovenia, which was socially distancing itself from foreigners, also distanced itself from Western Europe and aligned closer to the most intolerant post-socialist states. According to the international European Social Survey (Hafner-Fink 2004; Jowell 2003), typically, the evaluation of the immigration conditions for foreigners was very high (=strict) in Slovenian public opinion and, according to this criterion, Slovenia was placed together with Poland at the very top in terms of the extent of social distance (among the 13 European states in which the survey took place; see Hafner-Fink 2004); only Hungary advocated for even stricter immigration conditions. The mentioned immigration conditions included in the ESS that the respondents were asked to choose from included the following:

(a)   That they have a good education;
(b)   That their close relatives already live in the receiving country;
(c)   That they can speak the official language of the country (for Slovenia: "that they can speak Slovenian");
(d)   That they are trained for work skills needed by the country (Slovenia);
(e)   That they accept the way of life in the country (for Slovenia: "Slovenian way of life");
(f)   That they come from the Christian environment;
(g)   That they are white;
(h)   That they are wealthy.

In terms of strictness of criteria for immigration Slovenia is, as mentioned, at the very top among the post-socialist countries that are themselves at the top among the European countries, according to this criterion. Moreover, this refers to a time that saw the beginning of several years of economic conjuncture with the last economic crisis; let alone the refugee crisis, which was not even on the horizon yet. Even when we look at the evaluation by the Slovenian respondents according to these individual criteria from the above list that are the most xenophobic—namely, the criteria based on religious fundamentalism (*f*), racism (*g*), and elitism (*h*)—we obtain the same worrying picture:

-   With regard to the condition that an immigrant has to come from a Christian environment (*f*) Slovenia is placed high, in fourth place among 13 countries (right after Poland, Hungary, and Italy);
-   In the racist criterion that an immigrant has to be white (*g*), Slovenia is placed high, in third place (just after Hungary);
-   In the elitist criterion that an immigrant has to be rich (*h*), Slovenia ranks in third place (only preceded by Italy and Poland).

The aforementioned measurements in Slovenia and in Europe show that the two decades that followed saw four social shocks, which are known to have negatively influenced public opinion. The first shock was the last great economic crisis, which affected Slovenia a couple of years later (2009–2014); this was followed by the first great crisis in this millennium related to the rapid, sudden, unpredictable, completely unregulated, and mass arrival of refugees in 2015; the third shock was the COVID-19 pandemic; and the fourth shock was a militaristic one—the Russian invasion of Ukraine (2022)—and, a year later, the Israeli attack on Gaza. However, despite these events, public opinion in Slovenia has not become more negative to immigration, but rather it has slightly shifted toward a positive, more tolerant direction, as shown in Table 3.

**Table 3.** The attitude of Slovenian public opinion to immigrants and to quality of life[12].

| | Question | 2002 | 2016 | 2023 | Modality | |
|---|---|---|---|---|---|---|
| Migrations | Is it generally good or bad for the Slovenian economy that immigrants from other countries come to live here? | 4.29 | 3.99 | 5.50 | 0 = bad 10 = good | **Average** 11-grade scale |
| | Is cultural life in Slovenia generally threatened or enriched due to immigrants? | 5.21 | 4.73 | 5.25 | 0 = threatened 10 = enriched | |
| | Do immigrants from other countries make Slovenia become a worse or better country to live in? | 4.45 | 4.37 | 4.90 | 0 = worse 10 = better | |
| | To what extent should Slovenia allow the immigration of people with *similar national origin* to that of the majority population of Slovenia? | 62.9 | 73.0 | 84.1 | First two answers combined: it should allow to many + it should allow to some | **Percentage (%)** 4-grade scale |
| | To what extent should Slovenia allow the immigration of people with *different* national origin to that of the majority population of Slovenia? | 53.6 | 50.9 | 67.3 | | |
| | To what extent should Slovenia allow the immigration of people from *poor countries outside Europe*? | 53 | 51.7 | 66.5 | | |
| | Some think that the European Union should be even more unified within, others say that it is too unified already. What do you think? | 5.61[13] | 5.64 | 5.71 | 0 = unification went too far 10 = it is not unified enough | |
| Quality Of Life | On the whole, how happy would you say you are? | 6.93 | 7.47 | 7.78 | 0 = unhappy 10 = happy | **Average** 11-grade scale |
| | Do you think that most people would try to take advantage of you if they had the chance, or would most people try to act fairly? | 4.68 | 5.01 | 5.43 | 0 = most people would try to take advantage of me 10 = most people would try to act fairly | |
| | Do you think people are mostly willing to help others or are they mostly looking primarily after themselves? | 4.24 | 5.15 | 5.37 | 0 = they mostly look primarily after themselves 10 = they are mostly willing to help others | |
| | How safe do (would) you feel when you walk (would walk) alone in the evening in your neighbourhood? | 89.5 | 92.2 | 93.8 | First two positive answers combined: Very safe + safe | **Percentage (%)** 4-graded scale |
| Values | Obedience and respect of authority are the most important values that children should learn. | / | 72.0[14] | 75.4 | First two positive answers combined: Strongly agree + Agree | **Percentage (%)** 5-graded scale |
| | What Slovenia needs the most is loyalty to its leaders. | / | 35.0[15] | 35.1 | | |

Table 3 contains the data of three time cross-sections with the following characteristics. The characteristic of the first period (2002) was the economic conjuncture and the absence of political tensions; the second measurement (2016) reflects the reactions of the public to the aforementioned first refugee wave; and the most recent period (2023) is characterised by the outbreak of military hostilities in Europe and its neighbourhood, as well as the new increase in the arrival of refugees. As is shown in Table 3, in all the questions related to migration, public opinion in the second period became even more negative compared to the first; while in the last period the trend reversed toward the direction of a reduction of social distance toward migrants, which is now smaller than in both previous periods. Regarding the self-evaluated quality of life, in all the presented questions the share of satisfaction was increasing in all three periods. Only in values was there no progress whatsoever; in the question before the last one, an expressed majority share of those who favoured an authoritarian upbringing in the population notably increased, namely by 5 percent (or 3.4 of the percentage point), while in the last question about the "faithfulness to leaders" there were no changes, despite the fact that in 2022 great efforts were needed in Slovenia to change the previous authoritarian government, which was taking after the Hungarian autocrat Orban and the American Trump. In short, the Slovenian public is aware and appreciative of the quality of life that we still have at the moment, but at the same time, even though it traditionally leans towards authoritarian values (Dragoš 2016), public opinion is more favourable than unfavourable to migration. However, just the opposite is true for the political elites.

## 6. Attitude of the State to Refugees

In 2015, refugees arrived in Europe and Slovenia in huge numbers. The United Nations High Commissioner for Refugees pointed out that in only one year, i.e., from 2014 to 2015, the number of refugees in the world had increased from 60 to 65.3 million people; a number that has never been so high in history (UNHCR 2016). At the same time, in Slovenia 339 asylum applications were considered but only 17 were actually granted; while in a 20-year period Slovenia granted asylum to only 18 people per year (Petrovčič 2016).

How did European states react when they realised that the EU did not function? They reacted with arbitrary practices, whose only common characteristic was that the countries closed themselves to the outside and shifted the refugee burden to others. No EU member managed to avoid the repressive-isolationist fragmentation, with not a single one receiving refugees in its territory in even just one percent of its own population, regardless of the fact that this one percent would still be scandalously low even if it were realised. Therefore, on one side of this pan-European degradation there is the most multicultural and "open" Sweden, coming from the club of the richest states of the world; at the climax of the refugee crisis, it received the highest share of asylum seekers, considering the original population—with the share not even reaching 0.5 percent! In second place is Malta, which has an almost ten-times larger quota of asylum seekers than the richest country, Luxemburg. Although Malta consists of the smallest land mass in the EU, it also received a quota of asylum seekers that was almost ten times larger than that of France, which is the largest land formation in the EU. Third place goes to Austria and fourth goes to the wealthy Germany, which in this respect was the loudest and most influential at the time. The listed states are the only European "champions" in solidarity, provided it is measured in *tenths* of a percentage point. On the opposite pole, the countries that were the most closed to refugees were all former socialist countries—in which "solidarity" can be measured only in *thousandths* of a percentage point. The smallest numbers of refugees were accepted by Latvia, Hungary, Poland, and Slovenia, which received from 0.001 to 0.006 percent refugees with regard to the original population.[16] In short, in 2015, Europe obtained clear evidence that it does not handle the refugee issue effectively, and each member state reacts in its own way, all being distinctly isolationist in nature. At the time, this confusion was well illustrated by Šoltes, a Slovenian Member of the European Parliament with the following words:

"At the moment, difference in asylum legislation among EU members is so vast that in 2015, in some member countries, asylum applications from people from Iraq were approved only in 13 percent of cases, while the share of approved similar applications in other member countries could be very high, as much as 94 percent. Even this one case shows how necessary it is to harmonise national legislations". (Šoltes 2016)

After long negotiations at the European level, this conundrum lead to the proposal about refugee quotas being compulsory for each country, to provide a balanced reception of refugees. For example, this proposal would bind Germany to receiving 0.049 percent of refugees with regard to its original population; France 0.047 percent; Spain 0.041 percent; and Poland 0.031 percent. This is in contrast to, for example, Jordan where the share of refugees at that time would have been 38.5 percent; in Lebanon 26.3 percent; and Syria 5.5 percent (Dragoš 2016). What did Slovenia do in this regard? Like other post-socialist countries, which are the most xenophobic in Europe, it boycotted the proposed quotas. Therefore, quotas still do not work in Europe; no other solutions were offered either. Table 4 shows the most recent data on refugees, illustrating more clearly that the European problem is not the refugees but rather the policies acting against them.

**Table 4.** Absolute and relative shares of refugees in European countries in 2022 (Eurostat 2023a, 2023b).

| Countries | | (a) Persons with Temporary Protection | (b) Asylum Seekers | (c) Final Decisions of Asylum Applications (2011–2022) | (d) Original Population | % (a/d) | % (b/d) | % (c/d) |
|---|---|---|---|---|---|---|---|---|
| Largest | EU (27) | 4,332,250 | 955,525 | 251,780 | 448,400,000 | 0.97 | 0.21 | 0.0562 |
| | Poland | 1,567,905 | 9810 | 95 | 39,900,000 | 3.9 | 0.02 | 0.0002 |
| | Germany | 795,205 | 243,835 | 113,180 | 82,900,000 | 0.96 | 0.29 | 0.1365 |
| | France | 84,910 | 156,455 | 73,470 | 64,600,000 | 0.13 | 0.24 | 0.1137 |
| Smallest | Luxemburg | 5090 | 2460 | 75 | 640,064 | 0.79 | 0.38 | 0.0117 |
| | Malta | 1630 | 1320 | 105 | 518,536 | 0.31 | 0.25 | 0.0202 |
| | Iceland | 2305 | 4550 | 180 | 372,520 | 0.62 | 1.22 | 0.0483 |
| | Liechtenstein | 420 | 80 | 15 | 39,039 | 1.08 | 0.20 | 0.0384 |
| | *Slovenia* | *7480* | *6785* | *40* | *2,116,792* | *0.35* | *0.32* | *0.0019* |

We can see at the right side of Table 4 that a modest share of received refugees in European countries is plummeting when we compare it to the reception of persons with temporary protection and final decisions on asylum applications. While the European average in the former (a/d) is 0.97 percent of refugees with regard to the entire population of the receiving country, in final decisions (c/d) this share drops to only 0.056 percent. The last indicator (c/d) is 0.0019 percent. Slovenia belongs among one of the most closed countries. The real Slovenian capability of receiving refugees can be inferred from the fact that in the first year of its foundation Slovenia received, accommodated and took care of—and without any problems—the refugees coming from Bosnia and Herzegovina during the Yugoslav war. At that time, the mass influx of refugees to Slovenia represented as much as 3.5 percent of its population. Despite the pronounced instability in terms of foreign policy (the war in the neighbourhood) and critical economic conditions[17], there were no tensions between the refugees and Slovenian population, because at the time—as opposed to now—there was not a single political party that would fan hate-rhetoric towards the refugees. At the time we were a promising state; today, with the criterion of our migration policy, we are a state that seems to be losing its way.[18]

## 7. Conclusions

In relation to the attitude to refugees, the social function of the RCC in Slovenia is positive, mainly due to the merit of the Slovenia Caritas, which does good work. What is problematic is the connection between the highest-ranking bishops of the RCC with the right-wing political parties and the monopolistic position of the RCC in the religious "market", enabled by its enormous wealth and the systemically ensured financial and symbolic privileges provided by the state. The combination of both factors—the political and monopolistic ones—often takes the RCC to a markedly dysfunctional operation similar to the one occurring during the campaign for the building of the only mosque on Slovenian territory; at that time, the RCC supported the Islamophobic attitude typical of the conservative political parties, failed to distance itself from the inappropriate statements of some bishops, and has never apologised for this attitude. Therefore, it is likely that the RCC in Slovenia will repeat this pattern of behaviour in the future. This pattern is composed of four characteristics:

- The persistence of the RCC in conservative (pre-council) doctrinal positions and the advocacy of its own material interests even in situations where the broader social environment disapproves of them, leading to conflicts (in the triangle) between RCC actors, political parties, and the public.
- When asserting conservative positions and its own interests, the RCC never goes so far as to assume the role of an initiator who would trigger a dispute.
- Issues related to conservative positions and the interests of the RCC that become socially controversial in the Slovenian context are always brought to the public by right-leaning political parties when they come to power, and at that time, the RCC never misses the opportunity to support them and confront other actors around them.
- Regardless of the success or failure of disputes in which the RCC is involved in pursuing its own interests, and regardless of the consequences of such actions, the RCC in Slovenia never publicly regrets or apologises for anything.

So far, this pattern has been repeated by the RCC in Slovenia in response to all controversial issues, both regarding denationalisation and state funding of the RCC, as well as in the mentioned promotion of Islamophobia and also in the case of the attitude towards minority sexual identities and practices (Kuhar 2004; Vezjak 2006).

Will the same pattern be repeated in the case of refugees? The likelihood of such a scenario in the Slovenian context is more dependent on state policy and public opinion than on the proactive stance of the RCC. Slovenian public opinion is volatile in its attitude to refugees (similar to other European countries). Just after the foundation of the new state, the public was very tolerant of refugees (who were mainly Muslims); however, a decade later social distance increased, then slightly decreased after 2015. Recently, it has again showed signs of increase (Kos 2023), although we are one of the most closed European countries with the smallest share of refugees in absolute as well as relative terms. Key is the xenophobia of right-leaning political elites who, when they come to power, can easily intensify public resistance to refugees and in this case, the RCC is very likely to support them, following the described pattern of behaviour.

**Funding:** This research received no external funding.

**Data Availability Statement:** The availability of the data used in this article can be seen from the list of references.

**Conflicts of Interest:** The author declares no conflict of interest.

## Notes

[1] The most frequently mentioned are integration, stabilization, psychological, compensatory, interpretative, cultural, and critical functions (Beck 2009; Bruce 2003; Flere and Kerševan 1995; Furseth and Repstad 2006; Lavrič 2013; Smrke 2000; Turner 1991).

[2] Religion can be succinctly defined as the institutionalization of faith in the sense of the 22nd and 23rd definitions of religion, as provided by Stark and Bainbridge (1996, p. 326), where it is said: "*Religion* refers to systems of general compensators based on

[3] supernatural assumptions" (def. 22); "*Religious organizations* are social enterprises whose primary purpose is to create, maintain, and exchange supernaturally-based general compensators" (def. 23).

[3] The models mentioned by Beck (2009, pp. 173–204) are: the side-effect model (civilizing through the individualization of religions), the market model (commodity form of God), the model of religiously neutral constitutional state (as defined by Jürgen Habermas), the "world ethos" model (Hans Küng), and the methodological conversion model (Mahatma Gandhi).

[4] Before being founded as an independent state (in 1991), Slovenia was one of the six autonomous republics within the former socialist Yugoslavia.

[5] This version of denationalisation is unique in the world. The RCC's (once confiscated) property was restituted by the state of Slovenia in a 100 percent share and in nature, rather than in partial financial compensation as in all other post-socialist countries.

[6] If we add 55 registered religious organisations and the newly founded "spiritual" communities to the classical largest religious institutions active in the Slovenian territory before the foundation of the Slovenian state, there are altogether over 100 collectively organised actors. Despite this pluralisation of the religious space, the RCC is still considered by far the largest and strongest in Slovenia and maintains being approximately 24 times larger than the second largest (Islamic) religious community in terms of its membership (Lesjak and Lekić 2013, p. 155; for problems with the official registration of religious organisations see Lesjak and Črnič 2007).

[7] For example: "Many refugees and migrants who come to Europe from Northern Africa and the Middle East present a big challenge for all the members of the EU. All countries and citizens are called to solidarity/.../It is therefore important that our country and the EU provide opportunities to all refugees and migrants to legally obtain asylum for themselves and their families and integrate themselves in local communities. Slovenian bishops express support to all state institutions, the Slovenian Caritas and other humanitarian organisations that are or will be receiving refugees and asylum seekers." (SŠK 2021).

[8] In the economy, this is the largest affair of the RCC in the history of this institution; namely, the enormous investments of capital that the RCC acquired in Slovenia during denationalisation, then lost due to speculation, went bankrupt (800€ million), and also scammed the top ranks of the papacy in the Vatican in the process (Ivelja 2012; Ivelja and Modic 2012; Mekina 2012; Smrke 2014). For sexual violence in the RCC in Slovenia see (STA 2022; Petrovčič 2013; Mekina 2018).

[9] The opponents of the mosque rallied enough voices to call a referendum, which was then not called after all because it was prohibited by the Constitutional Court in 2004.

[10] Among the religiously determined population (including believers that do not belong to any organised religion) Catholics amount to over 86 percent (SURS 2002).

[11] It is often assumed that marginalised groups and lower social strata represent the origin of intolerance. In Slovenia, this does not apply. While there is a correlation between intolerant attitudes and some stratification indicators (such as education, financial situation; Toš and Vovk 2014) when intolerance spreads out, this, however, does not apply for its origin or beginning—the "merit" of the latter goes to the political elites. The beginnings of all major kinds of intolerance in Slovenia came from the top to the bottom, and the major indicator has always been the political elites (Pajnik and Fabijan 2022; Dragoš 2004, 2006, 2014, 2016). The same goes for our neighbouring country, Croatia (Erceg 2023).

[12] Source: (Hafner-Fink et al. 2023, pp. 18–21) (Adapted by: S. Dragoš).

[13] These data data are valid for 2006.

[14] The data are valid for 2020.

[15] See note 14 above.

[16] The calculation is based on official data on asylum seekers for 2015 (Eurostat 2016, p. 3). In contrast to economic and social migrations, asylum status "represents a universal human right that pertains to the individual—the refugee. The right to asylum is also enshrined in Article 14 of the Universal Declaration of Human Rights" (Samec 2005, p. 5).

[17] At that time, the Slovenian economy was not entirely compensated for the loss of the former Balkan market by re-orienting toward European markets, and also its GDP at the time was substantially smaller than it is today.

[18] Although Slovenia lacks a labour force (today we need from 40,000 to 50,000 foreign workers per year; Esih 2023), it fears refugees, despite its being considerably closed to them (Kos 2023).

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
