# Peer review of "The Catholic Church and Refugees in Slovenia"

_religions, doi:10.3390/rel15040387_

Round 1
Reviewer 1 Report
Comments and Suggestions for Authors
The manuscript discusses an interesting topic and provides some interesting results. However, the manuscript has certain shortcomings and it needs to be improved before publication. Firstly, the title is too general and it needs to be more precise (e.g. Catholic Church and attitudes toward migrants in Slovenia).
The introduction is too short and mainly focused on elaborating the following sections of the manuscript. The introduction lacks solid theoretical framework of the research and reflection on the previous researches related to the topic. Methodology of the research was also not clearly elaborated. The aim of the research as well as hypotheses/research questions need to be clearly stated.
The overall structure of the paper needs to be improved. The first part of the paper is focused on the position of the Roman Catholic Church in Slovenia and their attitudes toward migrants and the second part elaborates the results of the research on the attitudes of the Slovenians toward the migrants. Additionally, there is no strong and evident connection between those two parts. The author failed to establish a strong connection between belonging to Catholic Church and negative attitude toward migrants, since the results of the survey does not provide the information correlation between the religious beliefs of the respondents and their attitudes toward migrants. The manuscript also lacks solid discussion of the research results.
Comments on the Quality of English LanguageThe English language is good, it only needs minor language editing.
Author Response
I have considered all the comments from your review. In accordance with your instructions, I have made the following revisions:
- Modified the article title
- Expanded the introduction to include information on the basic methodological assumptions and the research objective
- Augmented the second chapter regarding the explanation of privileged funding for RCC in Slovenia
- Enhanced footnote 14 with additional justification of terminology
- Revised the final, concluding part of the article for a clearer result
- I have condensed the citation of references with my last name in the bibliography from the original nine to the four most essential ones.
All changes have been marked in the article with yellow highlighting.
Reviewer 2 Report
Comments and Suggestions for Authors
The document provides a comprehensive analysis of the role of the Roman-Catholic Church (RCC) in Slovenia in relation to migration and refugee issues. It presents data on the absolute and relative shares of refugees in European countries, highlighting Slovenia's relatively low reception of refugees compared to its population. The document also discusses the positive social function of the RCC, particularly through the Slovenian Caritas, in providing support to refugees. However, it also addresses the RCC's alignment with right-wing political parties and its susceptibility to Islamophobia, which has contributed to xenophobic sentiments in the country.
The thoroughness of the data analysis is evident in the presentation of statistical information, such as the absolute and relative shares of refugees in European countries, and the financial support provided by the Slovenian Government to Caritas for aid to migrants and refugees. The references cited, including Eurostat data and statements from RCC officials, contribute to the validity of the arguments presented. The document maintains coherence in its analysis, linking the data to the broader context of the RCC's role in migration and refugee issues in Slovenia.
In terms of objectivity and neutrality, the document presents a balanced view of the RCC's positive and negative impact on migration and refugee issues. It acknowledges the positive social function of the RCC, particularly through the Slovenian Caritas, while also addressing the problematic aspects, such as the RCC's alignment with right-wing political parties and its susceptibility to Islamophobia. The author's perspective is evident in the interpretation of the data, but the document maintains a factual and informative tone.
The article contributes to the existing literature on the topic by providing a detailed analysis of the RCC's role in migration and refugee issues in Slovenia. It sheds light on the complexities of the RCC's involvement, including its charitable activities and its alignment with right-wing political parties. The document's potential impact lies in enhancing the understanding of the multifaceted role of the RCC in shaping attitudes towards migration and refugees in Slovenia.
Overall, the article provides a valuable contribution to the understanding of the RCC's role in migration and refugee issues in Slovenia, offering a nuanced analysis of its impact and influence in this context.
There is an important gap regarding the Slovenian legal framework:
Both the international agreements Holy See- Slovenia and the group of domestic agreements/legal status of other religious entities should be studied: some of these minorities (Jewish, Protestants) are proportionally better financed than the Catholic entities. Only to contrast the myth echoed by the cited doctrine.
Also, the difference between asylum seekers (popularly included in refugees term) and irregular immigrants (illegal migration) should be explained at the beginning: humanitarian migration vs economic migration, or similar terms.
Hard criticism against RCC should be better founded, at least regarding wealth and Islamophobia (only based on the one-mosque issue). Something is already hinted in the text, by separating shepherds and herd, hierarchy and people, but more in deep reasoning would benefit such hard statements.
In short, the scope of the study is very interesting, but it lacks certain degree of objectivity. Such ideological results should be reoriented, or focused, to the correlation, or not, between norms on force, proportional financing and social awareness about (Catholic)Church-State (international) relations and other religious entities-government (domestic) relations. And then, respective assistance to migrants (not only refugees, or only them: it must be clarified).
Finally, a final reference list, separating primary and secondary sources would be better than only the final notes list.
Author Response
I have addressed the majority of the comments from your review. Here are the corrections:
- I provided a more detailed explanation of the terminology used (refugees, asylum seekers) both in the text and additionally in footnote 14.
- Expanded the introduction to include information on the basic methodological assumptions and the research objective.
- Augmented the second chapter regarding the explanation of RCC funding in Slovenia, making the normative status of this institution clearer, as well as addressing the normative issue regarding the constitutional provision on the separation of the state from the religious sphere.
- Revised the final, concluding part of the article for a clearer result.
- I have condensed the citation of references with my last name in the bibliography from the original nine to the four most essential ones.
However, I did not follow your accusation that the article is "not objective" or "ideological," nor your suggestion that in Slovenia: "some of these minorities (Jewish, Protestants) are proportionally better financed than the Catholic entities" - this is simply not true. There is a substantial body of scholarly literature, which I reference in the article, supporting the claim that the RCC is financially (and symbolically) privileged in Slovenia, as opposed to smaller religious organizations. There is no scholarly study proving the opposite. Regarding your criticism of bias and ideology, I find it challenging to address since the main issue lies in the interpretation of these qualifications. My understanding is that scientific "objectivity" hinges on data, not on an author's attempt to create the appearance of "impartiality." Similarly, with "ideology," the article would be ideological if it minimized critical assessments of the RCC and favored its praise (which already comprises about one-third of the article).
All changes have been marked in the article with yellow highlighting.

Round 2
Reviewer 1 Report
Comments and Suggestions for Authors
In comparison to the first version of the manuscript, this version has been improved notably, and the text is more coherent. The only remark is that the conclusion has a subjective tone, reflecting the author's personal opinion. It would be advisable to tone down the subjective note.
Comments on the Quality of English LanguageThe English language is mostly fine, but I suggest proofreading nevertheless.
Author Response
I would like to thank the reviewer for their comments and effort. Regarding the comment where it is stated: "The only remark is that the conclusion has a subjective tone, reflecting the author's personal opinion" - it is difficult for me to take action here, as attempting to "objectify" my subjective conclusions, as labeled by the reviewer, could potentially worsen the matter. I personally believe that there is nothing inherently wrong with subjective viewpoints as long as they meet two criteria simultaneously: that subjectivity does not conflict with the facts reflected in actual data; and secondly, that the subjective standpoint is recognized as such, meaning it is not disguised as "objectivity" and impartiality. Therefore, I believe it is better not to alter the formulated expressions.